

# Explicating genetic architecture governing nutritional quality in pigmented rice

Jebi Sudan[1], Uneeb Urwat[1], Asmat Farooq[1], Mohammad Maqbool Pakhtoon[1], Aaqif Zaffar[2], Zafir Ahmad Naik[2], Aneesa Batool[1], Saika Bashir[1], Madeeha Mansoor[1], Parvaze A. Sofi[2], Najeebul Ul Rehman Sofi[3], Asif B. Shikari[2], Mohd. Kamran Khan[4], Mohammad Anwar Hossain[5], Robert J. Henry[6] and Sajad Majeed Zargar[1]

[1] Proteomics Lab, Division of Plant Biotechnology, Sher-e-Kashmir University of Agricultural Sciences and Technology of Kashmir, Srinagar, Jammu and Kashmir, India
[2] Division of Genetics and Plant Breeding, Sher-e-Kashmir University of Agricultural Sciences and Technology of Kashmir (J&K), Srinagar, Jammu and Kashmir, India
[3] Mountain Research Centre for Field Crops, Sher-e-Kashmir University of Agricultural Sciences and Technology of Kashmir, Khudwani, Jammu and Kashmir, India
[4] Department of Soil Sciences and Plant Nutrition, Faculty of Agriculture, Selcuk University, Konya, Turkey
[5] Department of Genetics and Plant Breeding, Bangladesh Agricultural University, Mymensingh, Bangladesh
[6] Queensland Alliance for Agriculture and Food Innovation, Queensland University, Brisbane, Australia

Corresponding authors
Mohammad Anwar Hossain, anwargpb@bau.edu.bd
Sajad Majeed Zargar, smzargar@skuastkashmir.ac.in

## ABSTRACT

Rice is one of the most important staple plant foods that provide a major source of calories and nutrients for tackling the global hunger index especially in developing countries. In terms of nutritional profile, pigmented rice grains are favoured for their nutritional and health benefits. The pigmented rice varieties are rich sources of flavonoids, anthocyanin and proanthocyanidin that can be readily incorporated into diets to help address various lifestyle diseases. However, the cultivation of pigmented rice is limited due to low productivity and unfavourable cooking qualities. With the advances in genome sequencing, molecular breeding, gene expression analysis and multi-omics approaches, various attempts have been made to explore the genetic architecture of rice grain pigmentation. In this review, we have compiled the current state of knowledge of the genetic architecture and nutritional value of pigmentation in rice based upon the available experimental evidence. Future research areas that can help to deepen our understanding and help in harnessing the economic and health benefits of pigmented rice are also explored.

## INTRODUCTION

The sustainable food systems for ever-increasing human population include adequate food and also adequately balanced nutrient-rich food. An increased focus on foods that provide adequate nutrition has resulted from a deeper understanding of the role of nutrition in maintaining a healthy population, particularly in developing nations.

Rice is an important staple crop that feeds more than half of the world's population and is being cultivated on approximately 158 million hectares of land producing around 850

million tons of grains annually (*Krishnan et al., 2020*). Asia represents a major rice growing region amounting for about 85% of the total production, followed by Latin America and Africa. India, being the second largest producer of rice in the world contributes around134 million metric tonnes with productivity of 2.8 t/ha and in a total cultivated area of around 46 million hectares (https://www.statista.com/study/57630/agriculture-industry-in-india/). Rice is a good option for a healthy diet as it has no cholesterol, fat, or sodium and contains eight essential amino acids in a balanced proportion. Rice bran oil is rich in linoleic and oleic acid, which are essential for sustaining cell membranes and nervous system functioning (*Bhat et al., 2020*). Rice grains come in a variety of pigmentations, including yellow, green, brown, red, purple, and black. Pigmented rice (or coloured rice) has long been considered to have nutraceutical benefits resulting in ongoing production as niche rice in various parts of the world.

Pigmented landraces of rice have a higher content of total anthocyanin, total phenol and polyphenol which signifies high antioxidant potential (*Deng et al., 2013*). Due to their rich nutritional profile and high antioxidants, these rice types have the potential to boost human health by addressing a variety of metabolic disorders (Fig. 1). The consumption of coloured rice reduces oxidative stress while simultaneously increasing antioxidant capacity in animal models, and this may be linked to a lower risk of chronic diseases like cardiovascular disease, type 2 diabetes and some cancers (*Wongsa, 2021*). As a result of their health potential and widespread demand, several nations have evaluated their coloured rice and created newer types. Varieties of coloured rice, recognized for their nutritional advantages, have remained a primary crop in several regions of India (Fig. 1). Molecular breeding and biotechnology approaches are regularly used for increasing the nutritional components in pigmented rice as well as in transferring these quality traits to conventional white rice. However, there lies a gap between the research that is available in the form of specific publications and the thoughtful compiled information that is required to gain an overall understanding of the topic. Hence, the present review highlights information on the important nutritional components in pigmented rice grains along with the efforts made to increase the nutritional quality of both pigmented and white rice through various biotechnological approaches to address nutritional food security. The present review will serve as a comprehensive review for scholars, scientists and students working in the area of pigmented rice.

## SURVEY METHODOLOGY

For this article, we have conducted a rigorous literature review and collected information from research articles, review articles, book chapters, websites (http://www.statista.com/; https://agricoop.gov.in/en/Acts) and databases (http://rice.uga.edu/; https://www.gramene.org/; https://rapdb.dna.affrc.go.jp/). We have also used Google image for drawing a map of India (https://www.istockphoto.com/photos/india-map-outline). We excluded the studies having abstracts available only with no full-text articles.

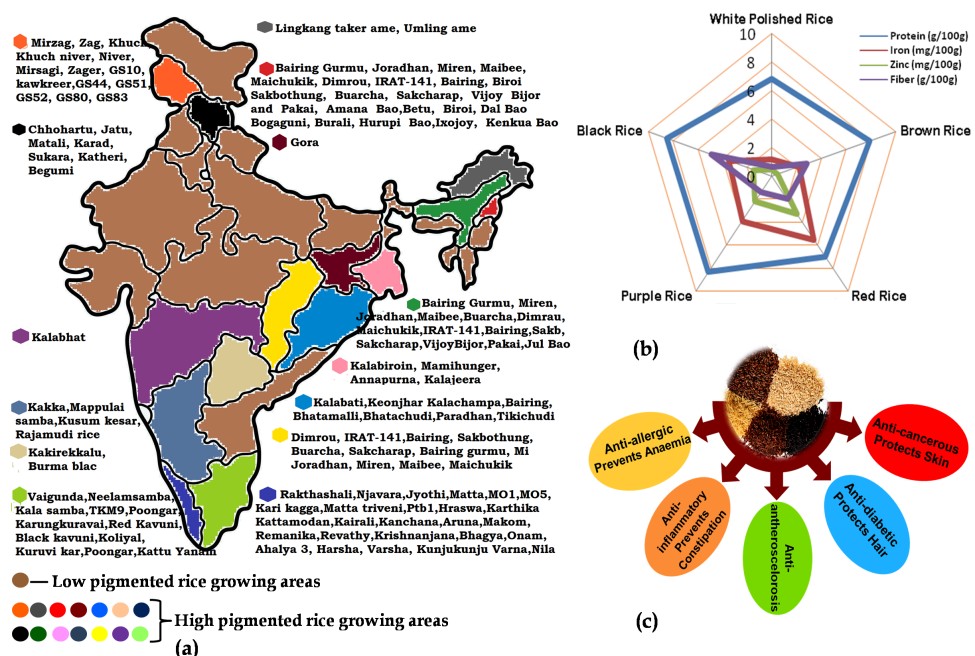

**Figure 1** The importance of pigmented rice in India with respect to the (A) varieties cultivated in different parts; (B) elemental content in different coloured-rice and (C) different nutraceutical benefits of pigmented rice.

# NUTRITIONAL COMPOSITION OF PIGMENTED RICE

## Protein and amino acids

Our bodies' healthy condition as well as tissue repair depends on the protein that we take from foods. Since rice is a staple food for a major part of the global population so rice protein is important in human nutrition. Rice protein concentrates and isolates are useful components in many food applications because they offer special nutritional qualities that set them apart from other cereals and legume proteins, such as a well-balanced amino acid profile and being easily digestible and non-allergic (*Sati & Singh, 2019*). Rice proteins are also excellent ingredients for baby food recipes due to their distinct nutritional and hypoallergenic qualities. Rice protein hydrolysates have the potential to improve food emulsion stability and serve as a natural antioxidant (*Rani, Pooja & Pal, 2018*). In rice, glutelins dominates seed storage proteins, making up to 60–80% of the total protein in the seed by weight. Compared to other grain seeds, that contains higher prolamine content as a proportion of total protein, they make up only roughly 20–30% of the protein in rice seeds (*Kawakatsu et al., 2008*).

Moreover, the amount and quality of protein is much better in pigmented rice as compared to non-pigmented rice. *Chen et al. (2022)* compared pigmented and white rice grains at five different stages of growth and found 230 differently abundant proteins associated with several metabolic activities. The pigmented grains were shown to have lower concentrations of proteins involved in signalling pathways, redox homeostasis, photosynthesis, nitrogen fixation, and tocopherol synthesis. In contrast, it was discovered

that the pigmented grain had higher levels of proteins necessary for the synthesis of sugars and flavonoids. Pigmented rices were also found to have a higher concentration of proteinogenic amino acids (histidine, threonine, valine, iso-leucine, methionine, phenyl alanine, lysine, proline, and tyrosine) and non-proteinogenic amino acids (glutamic acid, aspartic acid, asparagine, citrulline, and GABA) as compared to non-pigmented rice (*Samyor, Das & Deka, 2017*; *Kaur et al., 2018*).

Genetic engineering technology has substantially helped in improving cultivated crops through improvement of their essential amino acid and protein content. To raise the lysine content in cereals, lysine may be added at alternative codons during translation using a recombinant tRNA-lysine. Recombinant production of this tRNA in transgenic rice could greatly increase the amount of lysine in the seeds (*Wu, Chen & Folk, 2003*). (*Liu, Huang & Cai, 2013*) used an endosperm-specific promoter for expression of the lysine-rich protein (LRP) gene from *Psophocarpus tetragonolobus* that results in 30% more lysine in the transgenic rice seeds as compared to wild-type plants. Lysine biofortification in rice seeds was also achieved by over-expressing two endogenous lysine-rich histone proteins (RLRH1 and RLRH2) (*Wong, Liu & Sun, 2015*). Moreover, it was also found that increased lysine content also increases the threonine content in the grains through direct regulation (*Das, Adak & Lahiri Majumder, 2020*). The quantity of free tryptophan in the seeds increased significantly as a result of expression of an equivalent feedback-insensitive $\alpha$-subunit of the rice anthranilate synthase, but other important agronomical variables, including spikelet fertility, yield, and germination, are negatively correlated (*Wakasa et al., 2006*).

## Phytosterols, carotenoids and vitamins

Plant sterols and plant stanols are collectively referred to as "phytosterols". The natural phytosterol dietary intake varies from 150 to 450 mg per day, depending on a person's eating habits (*Ostlund, 2002*). Phytosterols have many beneficial effects such as anti-cancer activity, lowering blood levels of negative lipoproteins and cholesterol absorption (*Schaefer, 2002*). The three main and prevalent phytosterols found in human diet are $\beta$-sitosterol, campesterol, and stigmasterol. $\beta$-sitosterol is the most common phystosterol found in commercial rice cultivars, followed by campesterol, 15-avenasterol, and stigmasterol (*Zubair et al., 2012*). Three other sterols, including fucosterol, 24-methylene-ergosta-5-en-3b-ol, and 24-methylene-ergosta-7-en-3b-ol, are also present in the bran of the black rice variety "Riceberry" (*Suttiarporn et al., 2015*).

$\gamma$-oryzanol, is another class of phytosterols and is made up of a variety of phytosteryl ferulates, notably cycloartenyl ferulate, 24-methylenecycloartanyl ferulate, b-sitosteryl ferulate, and campesteryl ferulate. $\gamma$-oryzanol seems to be accumulated at a faster rate in pigmented grain compared to non-pigmented grain (*Chakuton, Puangpronpitag & Nakornriab, 2012*). Moreover, its value is found to be higher in black and purple rice as compared to red rice in Thai cultivars (*Sornkhwan, Chinvongamorn & Sansenya, 2022*).

The essential isoprenoid phytonutrients known as carotenoids, are produced in plastids, and are known to be lacking in rice endosperm. Provitamin A, found in carotenoids, are

also components that lower the risk of a number of diseases like cancer, heart disease, age-related muscular degeneration, immune system disorders, and certain other degenerative diseases (*Perera & Yen, 2007*; *Bollineni et al., 2014*. Lutein and zeaxanthin account for more than 90% of the carotenoids synthesised in rice but trace levels of other carotenes such as lycopenes and beta-carotene are also present (*Pereira-Caro et al., 2013*; *Melini & Acquistucci, 2017*). Most of these compounds are found in the bran, with milled rice containing little or no carotenoids (*Petroni et al., 2017*). The genetically variable feature of grain carotenoid concentration is highly associated with grain colour. White rice have extremely little carotenoid content, but red and black grains have much more (*Ashraf et al., 2017*; *Petroni et al., 2017*).

Vitamins constitute an important component of a balanced diet. Both tocopherols and tocotrienols (enriched sources of vitamin E) arefound in rice grains (*Zubair et al., 2012*). The most prevalent tocotrienols in rice are the $\beta$- and $\gamma$-tocotrienols (*Irakli et al., 2016*). According to *Gunaratne et al. (2013)*, red rice grains contain higher total levels of tocopherol and tocotrienol compared to white rice varieties. The quantity of tocopherol in the grain is, however, dramatically decreased by the processes of dehulling and milling (*Zubair et al., 2012*). Rice is deficient in vitamins A, C and D. However, brown rice has been found to contain significant amounts of vitamin B complex {thiamin (B1), riboflavin (B2), niacin (B3), pantothenic acid (B5), pyridoxine (B6), biotin (B7), folate (B9)} and E ( $\alpha$-tocopherol) (*Juliano & Bechtel, 1985*; *Samyor, Das & Deka, 2017*). Through molecular biology and genomic techniques, tremendous progress has been achieved in the genetic engineering of carotenoid production in plants during the past few decades. The whole collection of carotenoid biosynthesis pathway genes and related enzymes have been described. Metabolic engineering of the carotenoid biosynthesis pathway was carried out by marker assisted backcross breeding of two genes—phytoene synthase (*Zmpsy1*) from *Zea mays* and carotene desaturase (*Crtl*) from common soil bacterium *Pantoea ananatis* to create second generation transgenic rice, which accumulates phytoene, a crucial provitamin A intermediate (*Mallikarjuna Swamy et al., 2021*; *Biswas et al., 2021*). Due to their instability and degradation over time in long-term storage, folates in rice grain are less available. By genetic engineering a folate-binding protein, which increases the stability of folates by attaching to it over prolonged storage periods, biofortified high-folate rice grains were created that had 150 times more folate than wild rice (*Blancquaert et al., 2015*).

The mutant TNG71-GE brown rice variety was found to be richer in total tocopherol and tocotrienol than the wild-type. As a result, this mutant TNG71-GE rice variety could be used to produce a crop with high vitamin E content (*Jeng et al., 2012*). Later, it was also found that a single-point mutation of the giant embryo gene (*GE*) in Chao2-10 rice led to the development of a new mutant known as Shangshida No. 5. When compared to Chao2-10, Shangshida No. 5 brown rice contains more total vitamin E and $\alpha$-tocopherol (*Wang, Song & Li, 2013*). The overall amount of tocochromanols in rice was very slightly enhanced in transgenic rice created by constitutive overexpression of the *Arabidopsis thaliana* r-hydroxyphenylpyruvate dioxygenase (*HPPD*) gene (*Farré et al., 2012*). The elite Japanese rice cultivar Wuyujing 3 (WY3) provides transgenic brown rice with increased

quantities of $\alpha$-tocotrienol by both constitutive and endosperm-specific over expression of the *Arabidopsis g-TMT (AtTMT)* gene (*Zhang et al., 2013*).

## Flavonoids

Plants contain large amounts of flavonoids, which are secondary metabolites and play a significant role in plant development, pigmentation, UV protection, as well as in safeguarding the interaction with microorganisms (*Samanta & Das Gouranga, 2011*). Coloured flavonoids (flavanols, isoflavonoids, and flavones) are the pigments responsible for the colour of leaves, fruits, and flowers (*Yang et al., 2022*). Flavonoids play a crucial role in floral colours and fragrance, fruit pollinator attraction, and fruit dispersion (*Panche, Diwan & Chandra, 2016*). Due to the presence of flavonoids, terpenoids, steroids, and alkaloids, pigmented rice exhibits cytotoxic, anti-tumor, anti-inflammatory, antioxidant, and neuroprotective activities (*Goufo & Trindade, 2014*). In a recent study on thetotal flavonoid content (TFC) of rice, it was found that black and red rice had a higher TFC when compared to white rice varieties (*Chen et al., 2022*). Based on the amount of aromatic compounds, flavonoids can be classified into a wide range of groups such as chalcones, pro anthocyanidins, anthocyanins, flavones, flavonols, flavanones, and flavanonols (*Mbanjo et al., 2020*). However, proanthocyanidins and anthocyanins are the two primary flavonoids present in pigmented rice. Anthocyanins are responsible for the purple to blue coloration of such grains (*Zhang et al., 2023*). Consuming foods high in these substances may reduce inflammation and lower the risk of developing type-2 diabetes, cancer, and heart disease (*Rengasamy et al., 2019*). Eating foods high in anthocyanins on a regular basis also enhanced memory and overall brain health (*Henriques et al., 2020*).

Red and white rice grains lack anthocyanin (*Xiongsiyee et al., 2018*), although some red and brown rice accessions have low levels (*Ghasemzadeh et al., 2018*). The enzyme anthocyanin reductase converts unstable anthocyanidins into the colourless flavan-3-ols epiafzelechin, epicatechin, and epigallocatechin, and when these molecules are glycosylated, a broad variety of unique compounds are produced (*Kim, Kabir & Kabir, 2015*). After examining the transcription of eight flavonoid biosynthesis genes in various rice cultivars, it was found that pigmented variants had stronger expression of genes than non-pigmented forms (*Mbanjo et al., 2020*). At least two chalcone synthetase-encoding genes *CHS2* on chromosome 7 and *CHS1* on chromosome 11, help the production of flavanones in rice (*Cheng et al., 2014*). Proanthocyanidins are produced by three flavone 3-hydroxylases: *F3H-1* (on chromosome 4), *F3H-2* (on chromosome 10), and *F3H-3* (on chromosome 4) (*Park et al., 2016*). An effective method for researching genetic variants and genetic engineering in plants is the recombinant technology (*Mackon et al., 2021*). Understanding the function of several genes associated with anthocyanin biosynthesis and the development of anthocyanin in the endosperm has become possible due to analysis of the anthocyanin colouring mechanism in rice. The regulator of a rice prolamin gene was used to insert the maize *C1/R-S* regulatory genes into the white rice japonica cultivar Hwa-Young that resulted in the production of a wide range of flavonoid compounds (*Mackon et al., 2021*).

## Phenolics

Polyphenols constitute the most common secondary metabolites of plants and seem to be largely indigenous to the plant kingdom (*Dai, Dunn & Park, 2010*). They are essential for the plant's development, fertilization, and defence against viruses, parasites, and environmental conditions including light, cold, pollutants, and also impact the plant's colour (*Kabera et al., 2014*). Phenolic compounds benefit humans by reducing the risk of contracting chronic diseases, have a high antioxidant property and make significant contributions to the prevention of many oxidative stress-related diseases, including malignancy. Furthermore, there has been a lot of focus on identifying and synthesising phenolic compounds or extracts from diverse plants in the realms of health care and medicine (*Dai, Dunn & Park, 2010*).

Despite differences in the content of the grains, the amount of phenolic acid in brown, red, and black rice was found to be nearly the same. The different phenolic acids found in these pigmented rice varieties include ferulic acid, p-cumaric, sinapic, ferulic, and hydroxybenzoic acid. While ferulic, protocatechuic, and p-cumaric acids constituted the most prevalent common cell wall-bound phenolic acids, sinapic, ferulic (28%) and vanillic acids constituted the most significant soluble phenolic acids in black rice (*Blandino et al., 2022*). Black rice was found to possess higher protocatechuic and vanillic acids than brown rice (*Zaupa et al., 2015*; *Shao et al., 2018*). Using UV spectroscopy examination, the total phenolic content (TPC) of white, red and black rice varieties was determined. White rice types had TPCs much lower than those of black and red rice varieties (*Chen et al., 2022*). The genes involved in the biosynthesis of polyphenols interrelate with each other and have particular functions in the control of the polyphenols levels in rice grain (*Galland et al., 2014*). *OsCHS, OsCHI, OsF3H, OsF3′H, OsDFR*, and *OsANS* genes from brown rice have the ability to change the yellow seed coat of *Arabidopsis thaliana* to purplish. A yeast two-hybrid study revealed that, *OsCHS1, OsF3H, OsF3′H, OsDFR*, and *OsANS1* interact with one another directly (*Shih et al., 2008*). The rice mutant *Rcrd* turns red when *DFR* is introduced, demonstrating that Rd encodes the dihydroflavonol 4-reductase (*Furukawa et al., 2007*).

## Polysaccharides

Starch is the most abundant component which constitutes approximately 90% of rice grain. It is a polyglucan of two polymers, amylase (linear) and amylopectin (highly branched) with $\alpha$-1 →4-linked linear glucans and $\alpha$-1 →6-linked branches. Starch biosynthesis is a highly regulated process that requires synchronized activities between various enzymes such as starch synthase (SS), ADP-glucose pyro-phosphorylase (AGPase), starch branching enzyme (SBE) and de-branching enzyme (DBE). Studies have reported that starch synthesis in all higher plants and green algae and it is observed in higher plants, that the enzymes have undergone multi-sequential changes throughout the process of evolution (*Qu et al., 2018*). Conserved mutations in the enzymes- starch synthase SSIIIa and branching enzyme IIb (BEIIb) are helpful in breeding of highly resistant starch with more health benefits (*Bao, 2019*). Several efforts have been made to alter the expression and activities of starch biosynthetic enzyme by using various genetic as well as molecular approaches. Transgenic

rice plants have been beneficial for evaluating different functions of the genes responsible as rice is easily transformed and a single DNA construct can be used to produce a variety of transformed lines having different expression levels.

## Amylose and amylopectin content modification by alteration of single gene function

The modification of amylose content has been one among the most significant breeding objectives because it affects gelatinization and cooking qualities. A lower amylose content rice is usually selected as the milled grains are more appetizing than those with higher content (*Denardin et al., 2012*). Amylose is synthesized by the granule-bound starch synthases (GBSS) (Fig. 2). The amylose content in Japonica type rice (GBSSIb) in ss3a mutant was found elevated when it replaced indica type rice (GBSSIa) (*Crofts et al., 2012*). The regulation of *Os* GBSSI expression has been established in rice endosperm at the transcriptional as well as post transcriptional levels and that accordingly differentiates these two varieties of rice-indica and japonica in terms of amylose contents (*Liu, Huang & Cai, 2013*). The amylopectin content and its structure may be directly altered by the isozymes involved in its biosynthesis and the extent of such modification depends upon the isozyme's specificity. The change in amylopectin structure for example alters the starch to be highly resistant for gelatinization because of more and long double helices (*Miura et al., 2021*). There is also change in the X-ray diffraction patterns from wild-type to mutant-type because the amylopectin has longer external chains. Similarly the impact on amylopectin structure was seen when branching enzyme (BEIIb) activity in japonica rice was down regulated by the RNAi approach (*Tsuiki et al., 2016*; *Zhang et al., 2022*). The increase in amylopectin longer chains leads to an increase in the resistant starch content rather than an increase in amylose content (*Tsuiki et al., 2016*).

## Modification of amylose and amylopectin content by alteration in multiple gene functions

Multiple gene functions leads to intense modifications in the starch biosynthesis and thus, the phenotypes of these mutants exhibit variations. The changes are more extreme than predicted by the single gene mutants and the plant usually doesn't survive. For example, the triple mutants (ss1/ss2a/ss3a) of japonica rice type referred as null mutants becomes sterile which indicates the functional properties of each isozyme and importance of the presence of at least one of these (*SSI, SSIIa, and SSIIIa*) for starch biosynthesis in rice endosperm (*Fujita et al., 2011*). Other indirect effects are also observed such as the ADP glucose concentrations were found enhanced in the ss3a mutant as well as in ss3a/ss4b mutants when compared with wild type causing an increase in the Amylose content (*Fujita et al., 2007*). Similarly, the *BEIIb* and *BEI* gene inhibition results in the accumulation of amylose which leads to the different physico-chemical properties in the rice endosperm providing highly resistant starch. DNA editing with CRISPR/Cas9 twas found to be beneficial for specific editing of the *BEIIb* gene in rice for improved starch structure (*Baysal et al., 2016*).

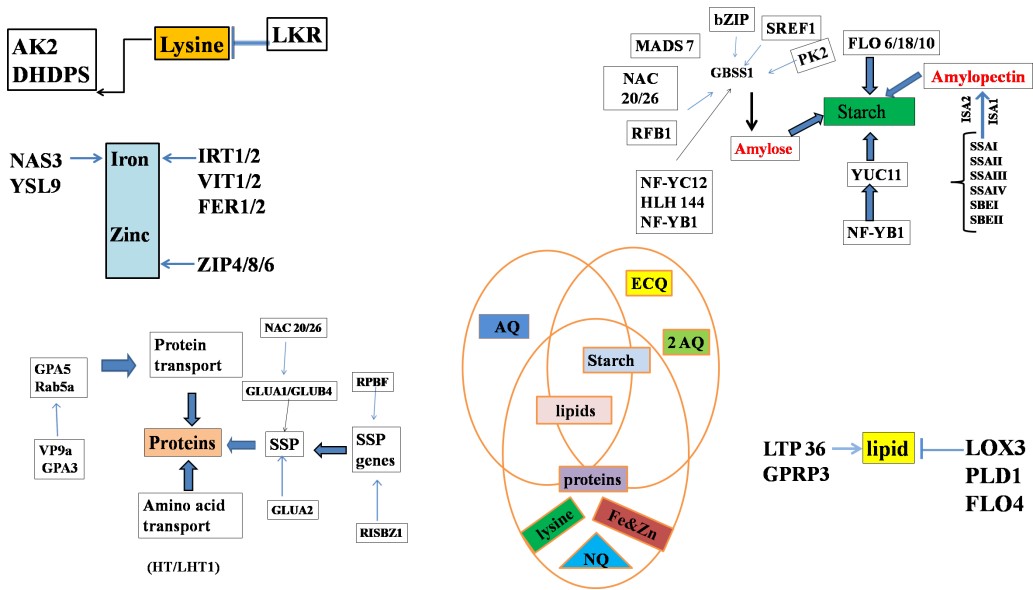

**Figure 2** Genes regulating starch biosynthesis in the endosperm through various transcriptional factors where, different abbreviations indicates amylose content (AC), eating & cooking quality (ECQ), nutritional quality (NQ) and appearance quality (AQ), granule-bound starch synthase (GBSS), soluble starch synthase (SS) and starch branching enzymes (SBE).

## Iron, zinc and micronutrients

Rice grain has trace amounts of several vital micronutrients, including zinc (Zn), magnesium (Mg), iron (Fe), copper (Cu), potassium (K), manganese (Mn) and calcium (Ca) (*Shao et al., 2018*). However, the coloured rice possess greater amounts of Zn, Fe, and Mn when compared with white rice (*Hurtada, Barrion & Nguyen-Orca, 2018*). Brown rice can deliver up to 75% of the optimal regular intake of zinc, copper, and iron while white rice only provides 37% (*Hashmi & Tianlin, 2016*). Red rice from North-East India has been observed to contain highest content of micronutrients (Al, K, Zn and Ca) than white rice (*Samyor, Deka & Das, 2016*). A recent survey found that two black rice variants from Korea, Heukjinjubyeo and Heukgwangbyeo possess greater levels of Ca and K in comparison to a white variety (*Hiemori, Koh & Mitchell, 2009*). It has been also observed that the ratio of micronutrients between coloured and non-coloured rice also differs. The important nutrients in polished rice samples were found to be in the order of K>Na>Mg>Zn>Fe>Mn>Cu>Cr, while in contrary to brown rice variants, the order is: K>Mg>Na>Mn>Zn>Fe>Cr>Cu (*Hashmi & Tianlin, 2016*). A survey was done in Northeast India about the mineral content in black, red and white rice varieties and a complementary investigation was conducted between mineral elements (Zn, Fe, Ca, Ni and Mn). The mineral elements like Fe, Ca, Ni, Mn, Zn in various varieties of pigmented rice were present in highest quantity than the white rice varieties (*Singh et al., 2022*). The growth conditions also affect the Zn and Fe concentrations in the purple rice varieties. Wetland conditions were found to be more advantageous than aerobic culture for producing purple

rice with vivid coloration and greater Zn (30 mg kg $^{-1}$) and Fe contents (15 mg kg $^{-1}$) (*Jaksomsak, Rerkasem & Prom-U-Thai, 2021*).

### β-carotene

Rice endosperm is not able to synthesize beta-carotene which acts as a precursor for vitamin A synthesis leading to deficiency in vitamin A especially in the developing countries where rice is the staple food and is the only way for fulfilling this nutritional requirement (*Das, Adak & Lahiri Majumder, 2020*). Vitamin A or β-carotene is a most important micronutrient for proper vision and development in humans and prevents many diseases such as night blindness, xerophthalmia and keratomalacia (*Klemm et al., 2010*). As rice endosperm is lacking β-carotene so it was necessary to make rice more nutritious by making it able to synthesize β-carotene (*Das, Adak & Lahiri Majumder, 2020*). In this context, an experiment was designed by *Burkhardt et al. (1997)* in which they found, in rice endosperm, geranyl geranyl diphosphate (GGPP), an important precursor for β-carotene biosynthesis making rice endosperm capable of β-carotene synthesis. The Rice Taipei 309 (japonica variety) was transformed with phytoene synthase from *Narcissus pseudonarcissus* by using micro projectile bombardment leading to β-carotene synthesis (*Burkhardt et al., 1997*). In another study the β-carotene content of Golden rice was enhanced ∼23-fold by replacing the daffodil psy gene with its homolog from maize and was named as Golden rice 2 (GR2) (*Mallikarjuna Swamy et al., 2021*) which is now a good source of vitamin A and might become a part of many breeding programs in Asia. Furthermore, β-carotene can be converted into the derivative astaxanthin which is keto-carotenoid and red in color with high antioxidant activity but most higher plants are unable to produce astaxanthin (*Ha et al., 2019*). Researchers genetically engineered *sPaCrtI* (phytoene desaturase), *sZmPSY1* (phytoene synthase), s*HpBHY* (b-carotene hydroxylase), and *sCrBKT* (b-carotene ketolase) genes to initiate the astaxanthin biosynthetic pathway to produce endospermic astaxanthin in rice grains (*Zhu et al., 2018*). *Tian et al. (2019)* reported that bioengineering of three chemically synthesized genes *i.e., tHMG1, ZmPSY1*, and *PaCRTI* in rice increased the endospermic carotenoid biosynthesis through the mevalonate route. This engineering of astaxanthin biosynthesis in rice endosperm converts Golden rice to aSTARice (*Zhu et al., 2018*). These improved rice genotypes contain one more gene for β-carotene hydroxylase that produces a red coloured Astaxanthin rice (aSTARice). Therefore, biofortification of rice through metabolic engineering could prove rice as a health promoting food and can be processed to produce dietary supplements (*Zhu et al., 2018*).

### Folate

Folates are a group of water-soluble B vitamins (B9), derived from most reduced folate form known as tetrahydrofolate (THF) contains three building blocks—the pteridine, *p*-aminobenzoate (*p*-ABA) and glutamate moieties (*Rébeillé et al., 2006*). Living organisms use folates as C1 donors and acceptors and are mainly involved in the biosynthesis and metabolism of nucleotides, amino acids and vitamin B5 (*Blancquaert et al., 2010*). However, only plants and micro-organisms can synthesize THF and its derivatives by *de novo* pathways. Therefore, humans are dependent on food to meet their daily
requirement of folates needed to regulate many physiological and molecular processes (*Blancquaert et al., 2014*) for their survival. Many plants like vegetables, pulses and fruits are loaded with folates but most staple crops such as rice which is consumed by 1/2 of the world's population; contain low levels of folate leads to folate deficiency especially in developing countries (*Dong et al., 2014*). To eradicate folate deficiency worldwide by biofortification of rice through metabolic engineering is a promising and cost-effective approach. Moreover, the concept of enhancing folate content by over expressing the folate biosynthesis genes/ metabolic engineering has been carried out in rice (*Strobbe & Van Der Straeten, 2017*) and is an interesting target for improvement (*Rébeillé et al., 2006*). The folate biosynthesis pathway in plants is a multi-step process occurs in three different subcellular compartments involves the conversion of chorismate (two-step process) by the action of ADC synthase (ADCS) into p-aminobenzoate (pABA) in plastids. Genes involved in folate biosynthesis such as *ADCS, GTPCHI, FPGS* and folate binding proteins (*FBP*) originated from different organisms has been genetically engineered and over expressed in rice to produce higher content of folates up to 100 fold (*Malik & Maqbool, 2020*) but further effective biofortification strategy is needed. The lists of genes engineered in rice as a single or in combination by different researchers to increase content of folates till now are mentioned in Table 1.

# PIGMENTATION IN RICE

Pigmented rice grains contain high levels of flavonoids, which are biosynthesized by two genes (*CHS1* and *CHS2*) encoding chalcone synthetase located on chromosomes 11 and 7, respectively (*Cheng et al., 2014*). Similarly, three flavone 3-hydroxylases, *F3H1, F3H2, and F3H3*, contribute to proanthocyanidin production in red rice grains (*Park et al., 2016*) and the two anthocyanin synthases are important for anthocyanins synthesis, ANS1 and ANS2 (*Shih et al., 2008*).

## Red and white pigmentation of rice grain

Rice grain colour was a major target during domestication and white rice was mostly selected, while most wild type rice is red. The colour is determined by the functional activities of different transcription factors. Two complementary genes, Rc and Rd (located on chromosomes 7 and 1, respectively), encode a basic helix-loop-helix (bHLH) transcription factor and are responsible for the red pericarp. RcRd genotypes produce red rice grain, while Rcrd genotypes produce brown rice grain (*Furukawa et al., 2007*). The Rc gene is also involved in rice grain dormancy and shattering. White variants have a loss-of-function mutation in the Rc allele (*Gross, Steffen & Olsen, 2010*).

## Purple rice pigmentation

The purple-pericarp formation in rice is determined by the gene *Kala4/OsB2/Pb* which produces anthocyanins (*Rahman et al., 2013*). Pigmentation variation is under polygenic control (*Ham et al., 2015*). The Pl locus on chromosome 4 has three alleles (*Plw, Pli*, and *Plj)*, each responsible for a different type of pigmentation. The wild type (*Plw*) produces anthocyanin in the aerial parts of the rice plant. Pl locus possesses the two genes- *OSB*-1

**Table 1  Genes engineered in rice seed for different nutrients and metabolites.**

| Nutrient/metabolite | Gene origin | Engineering approach | Genes transformed | Expression details/fold change in expression | References |
|---|---|---|---|---|---|
| Folate | *Arabidopsis thaliana* | Biosynthesis Single-gene | ADCS | (1/6)-fold upregulated | *Strobbe & Van Der Straeten (2017)* |
| | *Arabidopsis thaliana* | Biosynthesis Single | GTPCHI | 6.1-fold | |
| | *Arabidopsis thaliana* | Biosynthesis Single | HPPK/DHPS | 1.4-fold | |
| | *Triticum aestivum* | Biosynthesis Single | DHFS | 1.27-fold | |
| | *Arabidopsis thaliana* | Biosynthesis Multi-gene | GTPCHI + other biosynthesis genes | 6.1-fold | |
| | *Arabidopsis thaliana* | Biosynthesis Multi-gene | GTPCHI + ADCS | 100-fold | |
| | *Arabidopsis thaliana* | Polyglutamylation | FPGS | 1.45-fold | |
| | *Oryza sativa* | Polyglutamylation | FPGS | 4.7-fold | |
| | *Arabidopsis thaliana* | Polyglutamylation | GTPCHI + ADCS + FPGS | 100-fold | |
| | *Bos taurus* | Folate binding proteins | FBP | 6.2-fold | |
| | *Rattus norvegicus* | Folate binding proteins | GNMT | 8.8-fold | |
| | Arabidopsis thaliana (G + A) Bostaurus (FBP) | Folate binding proteins | GTPCHI + ADCS + FBP | 150-fold | |
| Lysine | *Arabidopsis thaliana* | Manipulation of lysine content | DHDPS | Feedback inhibition | *Das, Adak & Lahiri Majumder (2020)* |
| Carotenoids | *Zea mays* | Agrobacterium mediated DNA transfer | *OsLCYB* | 1.9-fold upregulated | *Tian et al. (2019)* |
| Carotenoids | *Pantoea ananatis* | Agrobacterium mediated DNA transfer | OsBCH2 | 1.1-fold upregulated | *Tian et al. (2019)* |
| | *Saccharomyces cerevisiae* | Agrobacterium mediated DNA transfer | OsPDS | 1.6-fold upregulated | *Tian et al. (2019)* |
| Vitamin A | *Narcissus pseudonarcissus* | Transformation by micro-projectile bombardment | pCPsyH | Expression observed in only the transformed plants. | *Burkhardt et al. (1997)* |
| Vitamin E | *Oryza sativa* | Single point mutation | Giant embryo gene (ge) | 2.2-fold upregulated | *Wang, Song & Li (2013)* |
| | *Arabidopsis thaliana* | Transformation by particle bombardment | PDS1 | upregulated | *Farré et al. (2012)* |
| Starch | *Solanum tuberosum* | Agrobacterium-mediated transformation | StGWD1 | 9-fold higher 6-phospho (6-P) monoesters and double amounts of 3-phospho (3-P) monoesters. | *Chen et al. (2006)* |
| | *Zea mays* | Agrobacterium-mediated transformation | OsSUS1-6 | Upregulatedstarch accumulation for improved grain filling. | *Fan et al. (2019)* |

Sudan et al. (2023), *PeerJ*, DOI 10.7717/peerj.15901

**Table 1** (*continued*)

| Nutrient/metabolite | Gene origin | Engineering approach | Genes transformed | Expression details/fold change in expression | References |
|---|---|---|---|---|---|
| | *Oryza sativa* | Map-based cloning-Mutation of T-DNA(gamma-radiated hybrid-rice) | SSIII and Waxy (Wx) | High in resistant starch (RS) | *Zhang et al. (2016)* |
| | *Arabidopsis thaliana* | RNA interference approach | Starch Excess 4 (SEX4) | improved bioethanol yield, with a 50% increase in ethanol production | *Huang et al. (2020)* |
| | *Oryza sativa* | RNA interference approach (antisenseWx gene) | Wx | amylose content in transgenic caryopsis was down-regulated | *Chen et al. (2006)* *Khandagale, Zanan & Nadaf (2018)* |
| | *Thermoan aerobacter ethanolicus* | Agrobacterium tumefaciens mediated transfer | APU (amylopullulanase) | Reduction of amylose, altered starch properties | *Liu et al. (2003)* |
| Nitrogen | *Arabidopsis thaliana* | Constitutive overexpression | MYB12, MYC, WD40 | Upregulated | *Wu et al. (2016)* |
| Phosphorus | *Arabidopsis thaliana* | Constitutive overexpression | OsMYB3R-2 | Upregulated | *Hwang, Lee & Tseng (2018)* |

& 2 encodes a helix loop helix transcription factor (*Sakamoto et al., 2001*). Purple colour is characterized by the two dominant genes Pb and Pp (*Ham et al., 2015*). A comparative genomics study in different rice lines by *Lachagari et al. (2019)* revealed important allelic variants in genes related to flavonoid synthesis, cytokinin glucosides, and betanidin degradation, and purple rice pigmentation.

## Black Rice Pigmentation

The *Kala1, Kala3*, and *Kala4* loci located on chromosomes 1, 3, and 4 express black pericarp traits in rice (*Maeda et al., 2014*). The pigments, mostly observed in the aleurone layers of black rice, are a mixture of anthocyanins and range from black to dark purple. Variations in the Kala4 promoter sequence mostly result in black rice grain phenotypes. *Kala4* encodes a Helix Loop Helix transcription factor which relates to *OSB2* gene responsible for the synthesis of anthocyanins (*Sakulsingharoj et al., 2014*). Genetic studies, including QTL mapping and GWAS analysis, have been used to understand the cause of rice pigmentation. Nine QTLs were reported for flour pigmentation in an inbred line population (*Tan et al., 2001*), and four QTLs were observed for red pigmentation (*Dong et al., 2014*). Twenty-one QTLs for variations in the composition and content of proanthocyanidins and anthocyanins were identified in a study by *Xu et al. (2017)*. Twenty-five marker trait associations for grain pigmentation were identified using GWAS (*Shao et al., 2011*). GWAS has been more useful than QTL mapping in determining the cause of rice pigmentation (*Korte & Farlow, 2013*). With the advance in GWAS, a total of 763 SNPs associated with pericarp pigmentation were reported by *Yang et al. (2018)* and some specific SNPs were also identified associated with Rc (*Butardo et al., 2017*).

## Marker trait association studies

Quantitative trait locus (QTL) is a part of DNA that affects quantitative trait and QTL mapping is a powerful and effective approach to analyse the chromosomal regions controlling quantitative traits for the marker-assisted selection (MAS) strategy in rice (*Hu et al., 2021*; *Islam et al., 2020*). Moreover, nutrient biofortification of rice by this method has been proved as a sustainable strategy to overcome mineral deficiencies (*Majumder, Datta & Datta, 2019*). Mineral accumulation in grain being a complex process and highly influenced by environmental factors has made breeding and early-generation phenotypic-based selections of biofortified rice varieties slow and less effective (*Sharma et al., 2020*). However, mapping major-effect QTLs by understanding genetics of grain mineral elements at the molecular level would be helpful for the rapid development of nutrient biofortification of rice varieties using marker-assisted breeding (MAB) (*Swamy et al., 2018*). To date, different mapping populations derived from biparental inter- or intra-subspecific and interspecific crosses have been used for identification of a large number of QTLs in rice genome which are mainly associated with mineral contents (*Wang et al., 2020*) and would be helpful in nutrient biofortification of rice varieties (*Hu et al., 2021*). In milled rice, 20 QTLs has been identified for P, K, Mg, Ca, Zn, Mn, and Cu contents (*Yu et al., 2015*) and in another study 51 QTLs in brown rice and 61 QTLs in rice straw was identified for different minerals (*Wang et al., 2020*) by using the RILs from the intra-subspecific

**Table 2  QTL studies showing different markers trait associations for nutrient enhancement in rice.**

| Grain Traits | QTL | Trait marker | References |
|---|---|---|---|
| Zinc (Zn) | qZN-5, qZN-7, qSZn2, qSZn12, qZn7, qZn3.1, qZn7.1, qZn7.2, qZn7.3, qZn12.1, qZn4, qZn6, qZn1-1, qZn12-1, qZn5-1, qZn8-1, qZn12-1, qZn2.1, qZn2.1, qZn3.1, qZn6.1, qZn6.2, qZn8.1, qZn11.1, qZn12.1, qZn12.2, qZn2.2, qZn8.3, qZn12.3, qZn3.1, qZn7, qZn8.3, qZn3-1, qZn1.1, qZn5.1, qZn9.1, qZn12.1, qZn1.1, qZn6.1, qZn6.2, qZn2-1, qZn2-2, qZn5, qZn10, qZn2.1, qZn3.1, qZn5.1, qZn5.2, qZn7.1, qZn9.1, qZn11.1, qZn 1.1, qZn2.1, qZn3.1, qZn3.2, qZn5.1, qZn6.1, qZn8.1, qZn8.2, qZn9.1, qZn12.1, QTL.Zn.4, QTL.Zn.5 | RMID2009463, RM2147095, RM2785595, RM6047367, RMID6006214, RM8832534, RMID11000778,RM12985052, RM13057679,RM34-RM237, RM7-RM517, RZ398-RM204 RM501-OsZip2, RM152, RM25-R1629, RM235-RM17, RM260-RM7102, RM551, RM413 | *Das, Adak & Lahiri Majumder (2020)*; *Islam et al. (2020)*, *Thangadurai et al. (2020)*, *Swamy et al. (2018)* |
| Iron (Fe) | qFE-1, qFE-9, qGFe4, qSFe1, qSFe12, qFe1, qFe3, qFe6, qFe2-1, qFe9-1, qFe4.1, qFe3.3, qFe7.3, Fe8.1, qFe12.2, qFe3-1, qFe9.1, qFe12.1, qFe1.1, qFe1.2, qFe6.1, qFe6.2, qFe5, qK6.1, qFe2.2, qFe3.1, qFe4.1., qFe6.1, qFe8.1, qFe11.2, qFe11.3, qFe12.1, | RM4743351, RM574-RM122, RM234-RM248, RM137-RM325A, RZ536-TEL3, RM270-RM17, RM260-RM7102, RM17-RM260, RM452, RM215 | *Das, Adak & Lahiri Majumder (2020)*; *Islam et al. (2020)*; *Thangadurai et al. (2020)*, *Swamy et al. (2018)* |
| Manganese (Mn) | qMn1-1, qMn2-1, qMn3-1, qMn10-1, qMn2.1, qMn2.1, qMn7.1, qMn1.1, qMn1.2, qMn3.1, qMn3.2, qMn4.1 | RM243-RM312, RM6367, RM227-R1925, RM214, RMID2009186, RMID2009463, RM7592793 | *Das, Adak & Lahiri Majumder (2020)*, *Mahender et al. (2016)*, *Swamy et al. (2018)* |

**Table 2** (*continued*)

| Grain Traits | QTL | Trait marker | References |
|---|---|---|---|
| Calcium | qCa1-1, qCa4-1, qCa5-1, qCa9-1, qCa10-1, qCa11-1, qCa12-1, qCa1.1, qCa1.1, qCa2.1, qCa2.1, qCa3.1, qCa3.2, | RM403585, RMID1013855, RMID2009186, RM2131264, RM2499734, RM2733626, RM598, RM5626-RM16, RM200-RM227, | *Das, Adak & Lahiri Majumder (2020)*, *Mahender et al. (2016)*, *Swamy et al. (2018)* |
| Magnesium (Mg) | qMg1-1, qMg3-1, qMg5-1, qMg9-1, qMg12-1, qMg3.1, qMg3.2, qMg5.1, qMg8.1, qMg9.1, qMg1.1, qMg7.1, qMg8.1, qMg11.1 | RM2499734, RM3460782, RM5522491, RM8892951, RM9886119,OSR 21, RM467, RM332, | *Das, Adak & Lahiri Majumder (2020)*, *Mahender et al. (2016)*, *Swamy et al. (2018)* |
| Phosphorus (P) | qP1-1, qP3-1, qP8-1, qP9-1, qP12-1, qP1.1, qP2.1, qP2.2, qP5.1, qP6.1, qP11.1, qP11.2 | RM,119519, RMID2009186, RM2181296, RM5430212, RMID6009257, RM3411, RM495, RM212, RM70-RM172, RM201, | *Das, Adak & Lahiri Majumder (2020)*, *Mahender et al. (2016)*, *Swamy et al. (2018)* |
| Potassium (K) | qK1-1 qK.1, qK4-1, qK8-1, qK9-1, qK2.1, qK4.1, qK4.2, qK5.1, qK9.1, qK3.1, qK3.2, qK3.3, qK4.1, qK5.1, | RM2094246, RM4285667, RM4668476, RM5430212, RM9858839, RM3572, RM5501, | *Das, Adak & Lahiri Majumder (2020)*, *Mahender et al. (2016)*, *Swamy et al. (2018)* |
| Boron (B) | qB2.1, qB3.1, qB4.1, qB4.2, qB10.1 | RMid2009186, RM2645329, RM4314701, | *Das, Adak & Lahiri Majumder (2020)*; *Swamy et al. (2018)* |
| Cobalt (Co) | qCo1.1, qCo3.1, qCo4.1, qCo12.1, qCo7.1, qCo10.1 | RM827062, RM2785595,RM4572241, RM12958034 | *Das, Adak & Lahiri Majumder (2020)*; *Swamy et al. (2018)* |
| Copper (Cu) | qCu3.1, qCu4.1, qCu4.2, qCu1.1, qCu1.2, qCu2.1, qCu6.1, qCu8.1 | RM3330180, RM4314701, RM4761773 | *Das, Adak & Lahiri Majumder (2020)*; *Swamy et al. (2018)* |
| Molybdenum (Mo) | qMo1.1, qMo1.2, qMo1.3, qMo2.1, qMo11.1, qMo12.1, qMo12.1, qMo12.1 | RM854218, RMID1014853, RM1191519, RM1725183, RMID11006537, RM13030749, RM13044018 | *Das, Adak & Lahiri Majumder (2020)*; *Swamy et al. (2018)* |
| Sodium (Na) | qNa1.1, qNa1.2, qNa7.1, qNa7.2, qNa10.1, qNa3.1, qNa11.1, qNa11.2 | RM267954, RM784044, RMID7003294, RM7962882, RM10635878 | *Das, Adak & Lahiri Majumder (2020)*; *Swamy et al. (2018)* |
| Phytic acid (PA) | qPA.12 | RM247-RM179 | *Thangadurai et al. (2020)* |

Sudan et al. (2023), *PeerJ*, DOI 10.7717/peerj.15901

**Table 2** (*continued*)

| Grain Traits | QTL | Trait marker | References |
|---|---|---|---|
| Grain protein content (GPC) | *qPC1, qPC2, qPC3, qPC6.1, qPC6.2, qPC8, qPC12.1, qPC1.1, qPC11.1, and qPC11.2, qPC-3, qPC-4, qPC-5, qPC-6 and qPC-10, qPr1 and qPr7, qPro-8, qPro-9 and qPro-10, qGPC1.1, qSGPC2.1 and qSGPC7.1,QTL.pro.1* | RM493-RM562, 1008-RM575, RM472-RM104, RM5619-RM1211, RM12532-RM555, RM251-RM282, RM190-RZ516, RM190-RZ516, RM270-C751, R1245-RM234, RM445-RM418, RM184-RM3229B, RM24934-RM25128, 1027-RM287, RM287-RM26755, RM5 | *Thangadurai et al. (2020)*, *Islam et al. (2020)*, *Mahender et al. (2016)* |
| Amino acid content (AAC) | *qAa1, qAa7, qAA.8, qAA.4, qAA.3, qAA.2, qAA.1, qAa9, qAA.10* | RM493-RM562, RM472-RM104, RM324-RM301, RM322-RM521, RM520-RM468, RM348-RM131, RM125-RM214, RM137-RM556, RM447-RM458, RM328-RM107, RM467-RM271 | *Thangadurai et al. (2020)* and *Mahender et al. (2016)* |
| **Amylose (amy)** | *QTL.amy.6, QTL.amy.7, QTL.amy.8, QTL.amy.11* | RM190, RM125, RM284, RM144 | *Islam et al. (2020)* |

cross Zhenshan 97/Miyang 46. *Garcia-Oliveira et al. (2009)* used introgression lines (ILs) from an interspecific cross of cultivar "Teqing" and Yunnan wild rice (*O. rufipogon*) and reported 31 QTLs in brown rice for P, K, Mg, Ca, Fe, Zn, Mn, and Cu contents. Some other researchers found 134 QTLS in brown rice for 16 minerals content by using both RILs and backcross introgression lines (BILs) from an inter-subspecific cross Lemont/Teqing (*Zhang et al., 2014*). *Descalsota-Empleo et al. (2019)* used two sets of doubled haploid (DH) lines from two inter-subspecific crosses IR64/IR69428 and BR29/IR75862 and reported 50 QTLs in milled rice for 13 minerals (P, K, Na, Mg, and Ca). Furthermore, *Du et al. (2013)* showed influence of environmental factors in the detection of QTL for grain mineral contents. In this study they selected brown rice grown in two different ecological environments namely Lingshui and Hangzhou and mapped 23 and 9 QTLs for seven mineral contents respectively and reported only two QTLs for the Mg content were found in both the environments simultaneously. Several other reports based on QTLs for mineral accumulation in rices have been detailed in Table 2.

## CONCLUSION AND FUTURE PROSPECTUS

Population growth and adverse global climatic changes negatively affect our food and nutritional securities which has resulted in hunger and malnutrition in our society. Rice being a staple food for half of the world can be a source of energy for our generations only if it is fortified with balanced nutrients. Pigmented rice as a food contain many bioactive compounds that display significant potential concerning a range of beneficial health effects like anti-cancerous, anti-allergic, anti-aging, anti-diabetic, and anti-obesity properties and include many medicinal properties like treating ulcer, fracture, burns, skin lesions, and many more. The extent of diseases faced today may be reduced significantly by simply replacing white rice with pigmented rice in our day to day diet. The present limitations of low productivity and palatability in pigmented rice can be solved by framing efficient breeding strategies along with use of multiomics approaches. Development of highly palatable and high yielding coloured rices will have a great impact in tackling various malnutrition concerns observed in rice eating countries and this can have great implications in attaining nutritional security.

## ACKNOWLEDGEMENTS

The authors are grateful to Vice Chancellor SKUAST-K (Prof. N.A. Ganai).

### Funding

This work was supported by the Department of Biotechnology, New Delhi, India (No.: BT/PR45619/NER/95/1937/2022). The funders had no role in study design, data collection and analysis, decision to publish, or preparation of the manuscript.

## Grant Disclosures

The following grant information was disclosed by the authors:
Department of Biotechnology, New Delhi, India: BT/PR45619/NER/95/1937/2022.

## Competing Interests

Mohammad Anwar Hossain is an Academic Editor for PeerJ.

## Author Contributions

- Jebi Sudan conceived and designed the experiments, performed the experiments, analyzed the data, prepared figures and/or tables, authored or reviewed drafts of the article, and approved the final draft.
- Uneeb Urwat performed the experiments, authored or reviewed drafts of the article, and approved the final draft.
- Asmat Farooq performed the experiments, authored or reviewed drafts of the article, and approved the final draft.
- Mohammad Maqbool Pakhtoon performed the experiments, authored or reviewed drafts of the article, and approved the final draft.
- Aaqif Zaffar performed the experiments, authored or reviewed drafts of the article, and approved the final draft.
- Zafir Ahmad Naik performed the experiments, authored or reviewed drafts of the article, and approved the final draft.
- Aneesa Batool performed the experiments, authored or reviewed drafts of the article, and approved the final draft.
- Saika Bashir conceived and designed the experiments, authored or reviewed drafts of the article, and approved the final draft.
- Madeeha Mansoor conceived and designed the experiments, authored or reviewed drafts of the article, and approved the final draft.
- Parvaze A. Sofi analyzed the data, authored or reviewed drafts of the article, and approved the final draft.
- Najeebul Ul Rehman Sofi analyzed the data, authored or reviewed drafts of the article, and approved the final draft.
- Asif B. Shikari conceived and designed the experiments, authored or reviewed drafts of the article, and approved the final draft.
- Mohd. Kamran Khan conceived and designed the experiments, authored or reviewed drafts of the article, and approved the final draft.
- Mohammad Anwar Hossain conceived and designed the experiments, authored or reviewed drafts of the article, and approved the final draft.
- Robert J. Henry conceived and designed the experiments, authored or reviewed drafts of the article, and approved the final draft.
- Sajad Majeed Zargar conceived and designed the experiments, prepared figures and/or tables, authored or reviewed drafts of the article, and approved the final draft.

## Data Availability

This is a literature review.

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
