# Peer review of "Explicating genetic architecture governing nutritional quality in pigmented rice"

_PeerJ, doi:10.7717/peerj.15901_

## Round 0.1 · original submission · Minor Revisions

Three independent experts in the subject assess the manuscript and find the work interesting, although they identified several concerns that the authors should address in the revision.

Reviewer 1 ·

Basic reporting

no comment

Experimental design

no comment

Validity of the findings

no comment

Additional comments

Authors have tried to compile the current state of knowledge of the genetic architecture and nutritional value of pigmentation in rice based upon the available experimental evidence.
In first para authors have written about Sustainable ways to increase production. Some lines must be added to support this.
Chemical formula or nomenclature used must be cross checked.
Use of lysine related content is good, please add few lines about lysine related omics.
Metals at first instance must not be abbreviated.
Name of genes and protein must be homogenized, somewhere they are italics and somewhere normal
At many places, space errors.
aSTARice...What is this? If any ,commercial variety, then details must be written.
For sustainable approach in plants, authors may cite
Multitrait Pseudomonas sp. isolated from the rhizosphere of Bergenia ciliata acts as a growth-promoting bioinoculant for plants, Front. Sustain. Food Syst. 7 (1097587)
Conclusion and future perspectives as per theme must be mentioned.
Subheading must be homogenized.............................somewhere first word is caps.
This section must be thoroughly edited----------Survey methodology
“For this article, we have done a rigorous literature review and collected information from research articles, review articles, book chapters, websites and databases. We collected agriculture stats from official websites, basics of nutritional composition of pigmented rice from reviews and the breeding/genetic engineering aspects were mostly taken from recent publications. We have also used Google image for drawing a map of India. We excluded the studies having abstracts available only with no full-text articles”
Detailed explanation, keywords used, which databases, websites etc. must be written....
“agriculture stats from official websites, basics of nutritional composition of pigmented rice from reviews and the breeding/genetic engineering aspects were mostly taken from recent publications”. THIS IS VERY VAGUE AND NON-SCIENTIFIC

“We excluded the studies having abstracts available only with no full-text articles”.............WHICH BIBLIOMETRIC DATABASES WERE TAKEN
Overall this article may be published with above considerations

Reviewer 2 ·

Basic reporting

no comment

Experimental design

no comment

Validity of the findings

no comment

Additional comments

In the manuscript entitled “Explicating genetic architecture governing the nutritional quality in pigmented rice”, the authors describe the overall nutritional components in the pigmented rice. They also addressed the efforts made to increase the nutritional quality of both pigmented and white rice through various biotechnological approaches to tackle nutritional food security. The pigmented rice varieties are rich sources of flavonoids, anthocyanin and proanthocyanidin that can be readily incorporated into diets to help address various lifestyle diseases. The findings of this review are well presented and contribute to the repertoire of studies conducted in order to understand the importance of pigmented rice. The authors collected huge information from various sources and discussed well about its nutritional values. Considering the quantity and quality of the work presented herein, I am pleased to recommend for its publication, after including the below raised points in the revised manuscript. I hope, the following comments might help the authors to improvise the overall manuscript.

Authors should check for typographical and grammatical errors in the manuscript.

Authors should cross-check all abbreviations in the manuscript. Initially, define in full name followed by abbreviation.

Figure 1 quality is not good for publication. Authors may improve the quality.

I would like that the conclusions fit the manuscript objectives, however the future prospective of the study should be added.

·

Basic reporting

The authors have compiled the information on research-based evidence of genetic architecture and nutritional value of pigmentation in rice. Also, the future strategies and outcomes have been discussed with emphasis on economic and health benefits. The authors summarized that the present limitations of low productivity and palatability in pigmented rice can be solved by framing efficient breeding strategies along with use of multiomics approaches. Development of highly palatable and high yielding colour rices will have a great impact in tackling various malnutrition concerns observed in rice-eating countries and this can have great implications in attaining nutritional securities.

Experimental design

The content is well designed and appropriate, But before acceptance and publication of this article authors must ensure that genes/ TFs name should be italics.

Also, do check that abbreviation has been fully abbreviated while using the first time in the whole manuscript.

Authors should include more references related to work from last five years(latest).

Validity of the findings

No Comments

---

## Round 0.2 · accepted · Accept

I greatly appreciate the authors' efforts in satisfactorily revising the manuscript. Your manuscript can be accepted with a little remark that the authors need to be aware of. I noticed that tables do not have gene names in italics. Please make sure that all of the gene and protein names in the manuscript adhere to the accepted notations.